# Temperature-Dependent Raman Studies of FAPbBr₃ and MAPbBr₃ Perovskites: Effect of Phase Transitions on Molecular Dynamics and Lattice Distortion

**Mirosław Mączka \* and Maciej Ptak**

Institute of Low Temperature and Structure Research, Polish Academy of Sciences, Okólna 2, 50-422 Wrocław, Poland; m.ptak@intibs.pl
\* Correspondence: m.maczka@intibs.pl

**Abstract:** Three-dimensional hybrid organic–inorganic lead halide perovskites are promising photovoltaic and light-emitting materials. A key phenomenon relevant for their optoelectronic applications is electron–phonon coupling. Since it can be strongly modified by structural deformation and changes in the dynamics of molecular cations, it is of great importance to study the temperature dependence of phonon properties of hybrid perovskites. In this work, temperature-dependent Raman scattering studies of FAPbBr₃ and MAPbBr₃ single crystals are reported in the 1800–22 cm$^{-1}$ and 300–90 K ranges. The Raman data of MAPbBr₃ showed clear anomalies near 236, 155 and 148 K, which were attributed to $Pm\bar{3}m{\rightarrow}I4/mcm{\rightarrow}P4/mmm$ (or $Imma$)$\rightarrow Pnma$ phase transitions. They also provided strong evidence that crystal structure of the phase stable in the 155–148 K range is very similar to structure of the $I4/mcm$ phase, not structure of the lowest-temperature $Pmna$ phase, as suggested in some reports. Therefore, the symmetry of this phase seems to be more likely $P4/mmm$ rather than $Imma$. An analysis of the temperature evolution of MAPbBr₃ Raman modes revealed that the phase transitions near 236 and 155 K are associated with weak distortion of the inorganic subnetwork and changes in the dynamics of MA$^+$ ions. Very pronounced changes in the lattice modes region and a narrowing of bands below 148 K indicated that the phase transition to the $Pnma$ phase is triggered by a freezing of MA$^+$ motions, which in turn leads to strong distortion of the inorganic subnetwork. Raman studies of FAPbBr₃ showed that this material behaves in a very different way than MAPbBr₃. First of all, the molecular dynamics of FA$^+$ cations are not frozen even in the lowest-temperature $Pnma$ phase. Moreover, the distortion of the inorganic subnetwork is small in the $Pnma$ phase. The observation of weak anomalies in the lattice modes region confirmed, however, that the two crystallographically resolved phase transitions ($Pm\bar{3}m{\rightarrow}P4/mbm$ near 260 K and $P4/mbm{\rightarrow}Pnma$ near 150 K) lead to weak distortion of the inorganic subnetwork. On the other hand, an analysis of FA$^+$ internal modes indicated that these transitions, as well as two other crystallographically unresolved transitions near 120 and 180 K, are triggered by a change of FA$^+$ dynamics.

**Keywords:** hybrid organic–inorganic perovskites; lead halides; Raman; phase transitions; methylammonium; formamidinium



## 1. Introduction

Hybrid lead halide perovskites crystallizing in various crystal structures, including zero-, one-, two- and three-dimensional (0D, 1D, 2D and 3D), have attracted a lot of interest in recent years due to their various functional properties [1–15]. Regarding 3D perovskites, the most famous ones are MAPbI₃ and FAPbI₃ (MA = methylammonium; FA = formamidinium) since they exhibit excellent photovoltaic properties with a power conversion efficiency exceeding 21% [7,8]. Bromide analogues are promising materials for tandem solar cells [9] and light-emitting diodes [1,2]. MAPbBr₃ was also shown to exhibit efficient multiphoton excited photoluminescence (PL) suitable for in vivo imaging

applications [10]. Very recently, the family of 3D lead halide perovskites was extended by two new members, MHyPbBr$_3$ and MHyPbCl$_3$ (MHy = methylhydrazinium) [13–15]. Similarly to the MA and FA analogues, MHyPbBr$_3$ and MHyPbCl$_3$ and their MHyPbBr$_{3-x}$Cl$_x$ solid solutions also exhibit PL properties and MHyPbBr$_3$ can also be classified as a very strong two-photon absorber [13–15]. It is worth adding that in contrast to the MA and FA analogues, which crystallize in centrosymmetric structures, the MHy counterparts adopt noncentrosymmetric and polar structures at room temperature (RT) [13–15]. As a result, these compounds exhibit second-harmonic generation (SHG) activity and MHyPbCl$_3$ was found to show quadratic nonlinear optical (NLO) switching behaviour [13–15].

It is well-known that optoelectronic properties of lead halide perovskites are affected by electron–phonon coupling [16]. Since phonon properties and thus the strength of electron–phonon coupling may be strongly affected by distortion of the Pb-X framework and ordering–disordering of organic cations due to structural phase transitions, it is of great interest to understand the effect of temperature on lattice dynamics and structural changes in this family of compounds. One of the methods that is most widely used in such studies is Raman spectroscopy [11–13,17–22].

X-ray diffraction data showed that MAPbBr$_3$ exhibits three phase transitions at 236, 153 and 144 K, which lead to a decrease of symmetry from cubic ($Pm\bar{3}m$) to tetragonal I ($I4/mcm$), tetragonal II ($P4/mmm$) and orthorhombic ($Pnma$), respectively [22–25]. However, very recent X-ray diffraction data suggested that the intermediate phase, ascribed previously to the tetragonal phase II, has incommensurately modulated structure with the average space group $Imma$ [26]. X-ray and neutron diffraction data indicated that MA$^+$ cations are disordered in the $Pm\bar{3}m$ and $I4/mcm$ phases but ordered in the orthorhombic $Pnma$. In the case of FAPbBr$_3$, X-ray diffraction revealed the presence of two phase transitions at 266 and 156 K, which lead to a decrease of crystal symmetry from cubic ($Pm\bar{3}m$) to tetragonal ($P4/mbm$) and orthorhombic ($Pnma$), respectively [27]. Three additional phase transitions, not resolved crystallographically, were observed at 182, 161 and 120 K as clear and narrow peaks in heat capacity studies [11]. The mechanism of these phase transitions remains elusive, but it was suggested that they must relate to changes of the FA$^+$ dynamics [27].

There are a few reports on temperature-dependent Raman spectra of MAPbBr$_3$ and FAPbBr$_3$ [11,19–21]. The deepest analysis of MAPbBr$_3$ Raman data was presented by Leguy et al. but the observed wavenumbers were reported for the $Pnma$ phase only and wavenumber vs. temperature plots were not shown [21]. Such plots were shown by Nakada et al. and Simenas et al. but only for a few selected modes observed above 300 cm$^{-1}$ [11,20]. Furthermore, Raman data reported by Nakada et al. were recorded for a powdered sample [20]. Regarding FAPbBr$_3$, there are only two literature reports on temperature-dependent Raman studies of this material [11,19]. However, the Raman spectra were presented in the 1020–420 cm$^{-1}$ range only [11] or the low-temperature Raman spectra were obscured by a strange broad and intense feature near 500 cm$^{-1}$ [19]. Furthermore, Raman wavenumbers were listed for the RT phase only [19] and the RT spectrum presented in [19] showed very different bands in the lattice modes region than the Raman spectrum reported in [22].

Since phonon properties of all phases and phase transition mechanisms are still not fully understood, especially in the case of FAPbBr$_3$, we decided to perform very detailed temperature-dependent Raman studies of MAPbBr$_3$ and FAPbBr$_3$ single crystals using small temperature steps and a broad wavenumber range (1800–22 cm$^{-1}$). We show that these data provide a deeper insight into temperature-driven structural changes in these compounds, especially regarding the role of MA$^+$ and FA$^+$ cation dynamics in the phase transition mechanisms.

## 2. Materials and Methods

### 2.1. Materials and Synthesis

Single crystals of MAPbBr$_3$ (FAPbBr$_3$) with dimensions up to 10 mm were grown from mixtures containing PbBr$_2$, acetonitrile, DMSO, HBr and a methanol solution of methylamine (formamidinium acetate) by antisolvent vapour-assisted crystallization using methyl acetate as the antisolvent (further details can be found in [11,24]).

### 2.2. Raman Spectroscopy

The Raman spectra of MAPbBr$_3$ and FAPbBr$_3$ single crystals in the 1800–100 cm$^{-1}$ range were measured using a Renishaw inVia Raman spectrometer (Renishaw, Wotton-under-Edge, UK), equipped with a confocal DM2500 Leica optical microscope and a thermoelectrically cooled CCD as a diode laser operating at 830 nm. A 20 × 0.4 microscope magnification lens was used, the size of the studied crystals was less than one mm and the laser spot diameter was about 0.75 μm. No sample inhomogeneity was observed under the microscope. The same spectrometer was used to record Raman spectra in the low-wavenumber range (250–22 cm$^{-1}$) but in this case an eclipse filter was employed. The temperature was controlled using a THMS600 stage (Linkam, Tadworth, UK) and the spectral resolution was 2 cm$^{-1}$.

## 3. Results and Discussion

### 3.1. Temperature-Dependent Raman Study of MAPbBr$_3$

The temperature-dependent Raman spectra of a MAPbBr$_3$ single crystal are presented in Figures 1 and 2, whereas plots of the wavenumbers and full width at half-maximum (FWHM) values vs. temperature are presented in Figures 3 and 4. The observed modes are listed in Table S1 together with the assignment based on previous Raman scattering studies of MAPbBr$_3$ [20,21]. It should be noted that the mode near 322 cm$^{-1}$ was assigned by Leguy et al. to the C–N torsion but we adopted here the assignment proposed by Nakada et al., i.e., we assigned this band to the MA-cage mode since this mode is not a pure torsion, as evidenced by the fact that it is strongly sensitive to the type of halogen anion [20]. The most intense modes, observed in the 35–81 cm$^{-1}$ range at 90 K, can be assigned to octahedra twist and octahedra distortion (or alternatively to octahedra librations and Pb–Br bending modes). The remaining lattice modes, observed in the 90–141 cm$^{-1}$ range, involve librational and translational modes of MA$^+$ but since these modes couple to the cage modes [21], we assigned them to coupled L(MA$^+$), T'(MA$^+$) and Pb–Br stretching modes (Table S1). It is worth noting that the intensity of these lattice modes is at least one order of magnitude stronger that the intensity of the internal modes, confirming the contribution of Pb–Br vibrations to these modes.

In the 300–230 K range (phase $Pm\bar{3}m$), almost all modes shift to higher wavenumbers with a decrease of the temperature (Figure 3). The only exceptions are the 1430 and 58 cm$^{-1}$ modes. The $Pm\bar{3}m \rightarrow I4/mcm$ phase transition, which occurs at 236 K, is visible as a slight step-like downshift of the ρ(NH$_3$) + ρ(CH$_3$) mode near 918 cm$^{-1}$ and a change of slope of FWHM vs. temperature for the 1250 and 1430 cm$^{-1}$ modes (Figure 3a,c). A weak wavenumber downshift is also observed for the 58 cm$^{-1}$ lattice mode (Figure 4a). This mode also shows a clear discontinuous decrease of FWHM (Figure 4b). These changes indicate that the phase transition leads to a weak slowing down of the reorientational motions of MA$^+$ cations and changes of PbBr$_6$ tilts. This conclusion is consistent with a disorder of MA$^+$ cations in the $I4/mcm$ phase.

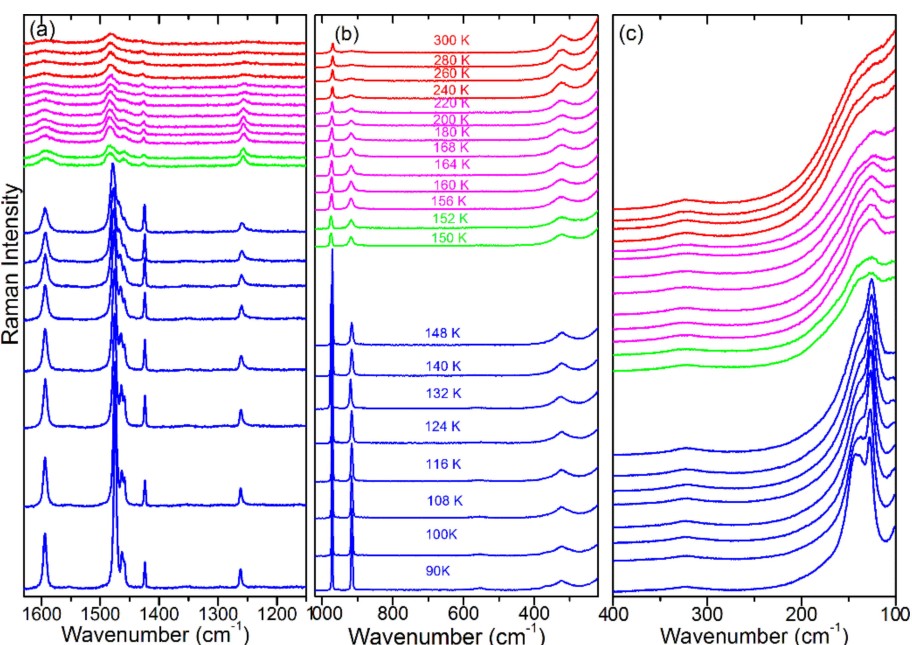

**Figure 1.** Temperature-dependent Raman spectra in the (**a**) 1630–1150 cm$^{-1}$, (**b**) 1020–220 cm$^{-1}$ and (**c**) 400–100 cm$^{-1}$ range. Red, magenta, green and blue colours correspond to the $Pm\overline{3}m$, $I4/mcm$, $P4/mmm$ (or *Imma*) and *Pnma* phases, respectively.

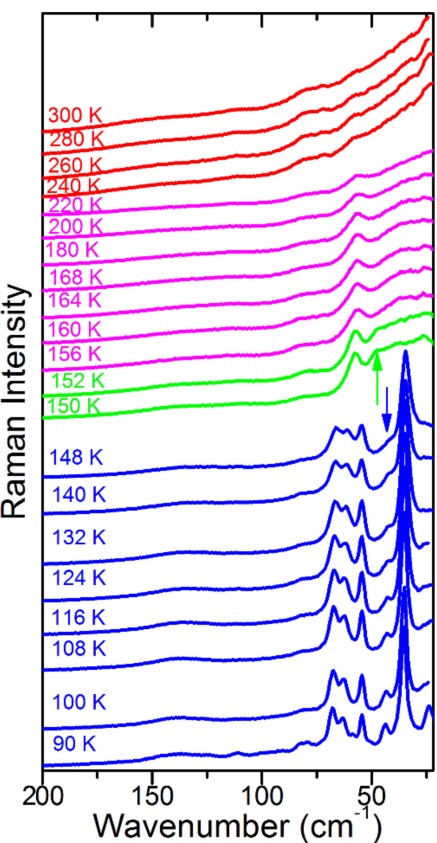

**Figure 2.** Temperature-dependent Raman spectra in the 200–22 cm$^{-1}$ range. Red, magenta, green and blue colours correspond to the $Pm\overline{3}m$, $I4/mcm$, $P4/mmm$ (or *Imma*) and *Pnma* phases, respectively. Green and blue arrows indicate bands, which appear in the low-temperature phases at about 47 and 43 cm$^{-1}$, respectively.

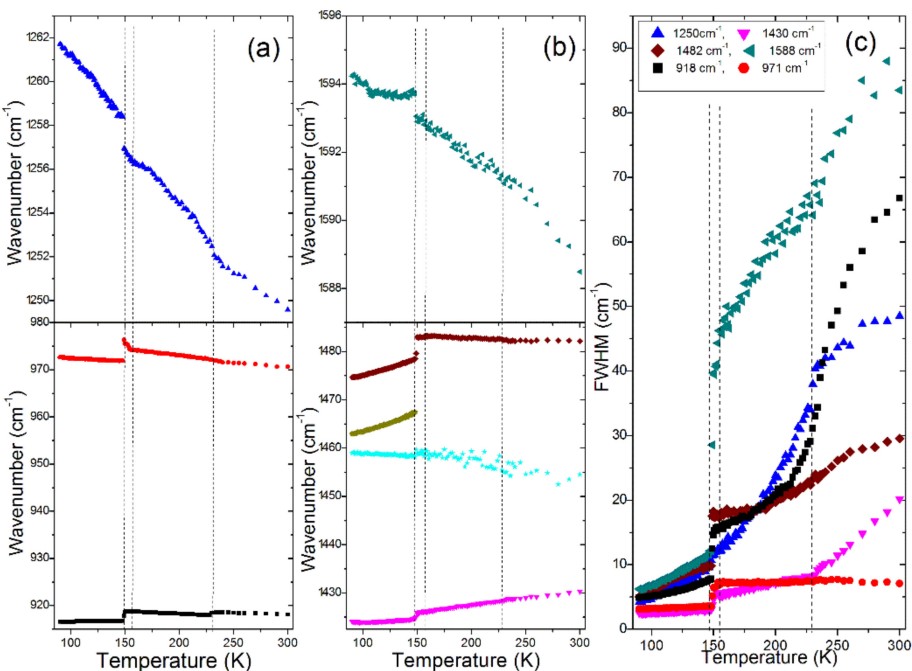

**Figure 3.** Plots of Raman (**a**,**b**) wavenumbers and (**c**) FWHM for internal modes of a MAPbBr$_3$ single crystal. The same color in (**c**) and (**a**,**b**) denotes FWHM and wavenumber data for the same mode.

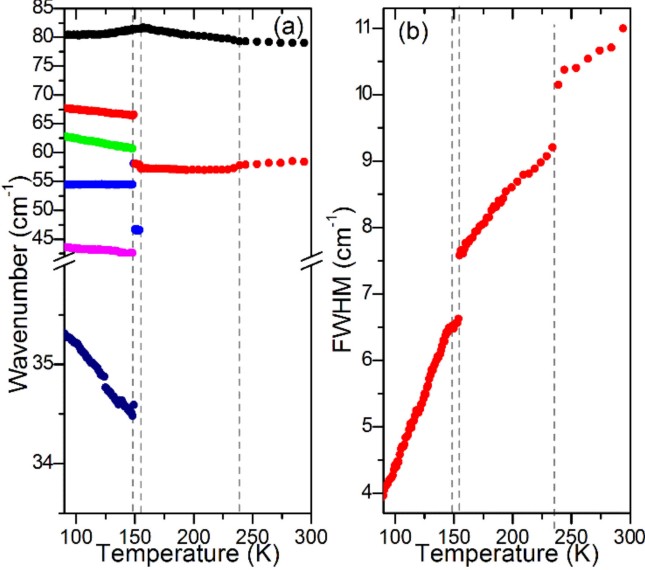

**Figure 4.** Plots of Raman (**a**) wavenumbers and (**b**) FWHM for lattice modes of a MAPbBr$_3$ single crystal. The same color in (**a**,**b**) denotes data for the same Raman mode.

On further cooling, the Raman spectra do not show any clear changes down to 156 K, except a narrowing of bands, which indicates a slowing down of molecular motions. Transition to a new phase becomes evident at 155 K. Firstly, the 971 cm$^{-1}$ band shows a clear change of wavenumber slope vs. temperature (Figure 3a). Secondly, a new band appears at 47 cm$^{-1}$ (Figure 2) and the 58 cm$^{-1}$ band exhibits a sudden narrowing (Figures 2 and 4b). However, the overall shape of the spectrum is very similar to that observed at 156 K. The Raman spectra indicate, therefore, that although the phase transition is associated with a further slowing down of the MA$^+$ reorientational motions and a change in the tilts of PbBr$_6$ octahedra, the structure of the new phase is very similar to that of the tetragonal phase I. In other words, we do not see any Raman evidence that could indicate

a lowering of the crystal symmetry to *Imma*, as recently reported [26], i.e., the *P4/mmm* symmetry of this phase seems to be more likely.

The third phase transition is observed at 148 K. It leads to a sudden decrease of FWHM of all bands (Figures 3c and 4b), especially pronounced for the 1588, 1482 and 918 $cm^{-1}$ bands that involve vibrations of the $NH_3$ group (Table S1). Furthermore, internal modes exhibit sudden shifts, $\delta_{as}(CH_3)$ and MA-cage modes split into doublets, the number of lattice modes increases to six and the intensity of the majority of the lattice and internal modes strongly increase (Figures 1–4, Table S1). The Raman spectra prove, therefore, that this phase transition is triggered by an ordering of the $MA^+$ cations, associated with a strong distortion of the inorganic subnetwork and a lowering of the crystal symmetry, i.e., the phase transition mechanism is of an order–disorder type.

It is important to compare our results with the previous data reported in the literature for a single crystal [21] and a polycrystalline pellet [20]. Previous studies of Leguy et al. showed more internal modes than observed by Nakada et al. Since our spectra show the same bands as reported by Nakada et al., we suppose that some weak bands reported by Leguy et al. appeared due to impurities or surface defects. The temperature evolution of the 1482, 971 and 918 $cm^{-1}$ modes is also in good agreement with that reported by Nakada et al. [20]. However, the very small temperature steps used in our experiment allowed for a much more precise observation of the changes induced by the phase transitions, especially in the narrow temperature range corresponding to the *P4/mmm* (or *Immm*) phase. Furthermore, we could also observe a large hardening and very pronounced narrowing of the 1588 $cm^{-1}$ band on cooling the crystal, not clearly seen by Nakade et al., and monitor the temperature dependence of the 1250 and 1430 $cm^{-1}$ modes, not reported in the literature.

In the lattice modes region, a similar number of bands was observed in all studies. However, a large discrepancy can be noticed as far as the Raman modes of the intermediate *P4/mmm* (or *Immm*) phase are concerned. First of all, Leguy et al. showed that the most intense and lowest wavenumber Raman band (35 $cm^{-1}$ in our case) remains the most intense in this phase, i.e., the Raman spectrum of this phase is similar to the Raman spectra of the lowest-temperature *Pnma* phase [21]. On the other hand, Nakada et al. showed that the Raman spectrum of this phase was similar to the Raman spectra of the tetragonal phase II [20]. Our data very clearly show that the latter case is correct.

The presence of structural phase transitions can affect light-emitting application of $MAPbBr_3$. First of all, a change in the lattice distortion should affect the bandgap energy and thus the PL colour [25,28]. Since we observed very weak (strong) changes in the distortion of the inorganic subnetwork at the $Pm\bar{3}m{\rightarrow}I4/mcm{\rightarrow}P4/mmm$ ($P4/mmm{\rightarrow}Pnma$) phase transitions, these transitions should weakly (significantly) affect the PL colour. Indeed, literature data show that as the temperature decreases, the PL peak does not show any sudden changes at the higher-temperature phase transitions, but it exhibits a sudden shift to higher energy at the $P4/mmm{\rightarrow}Pnma$ phase transition [25,28]. The optoelectronic properties of hybrid perovskites should also be affected by the change in the molecular dynamics. In particular, the disordering of organic cations may significantly increase the dielectric screening in hybrid perovskites leading to an increase in the static dielectric constant [25]. This effect is known to reduce the nonradiative recombination of excitons, which results in an increased PL during the order–disorder transformation [25]. Our Raman data show that $MA^+$ cations, fully ordered in the orthorhombic phase, become partially disordered in the *P4/mmm* phase. Therefore, this phase transition should affect the PL intensity. This behaviour is indeed observed since the PL intensity suddenly increases when going from the *Pnma* to *P4/mmm* phase [25]. A change in the lattice dynamics due to a phase transition can also affect the exciton binding energy. In the case of MA-based lead halides, this energy decreases when going from the orthorhombic to the tetragonal phase [29]. A lower exciton binding energy is beneficial for photovoltaic applications.

### 3.2. Temperature-Dependent Raman Study of FAPbBr₃

Temperature-dependent Raman spectra of a FAPbBr$_3$ single crystal are presented in Figures 5 and 6, whereas plots of the wavenumbers and FWHM values vs. temperature are presented in Figure 7. The observed modes are listed in Table S2 together with the assignment based on previous Raman scattering studies of FAPbBr$_3$ and a DFT calculation of the FA$^+$ cation [19,22,30]. We have, however, modified assignment of the modes observed near 115 cm$^{-1}$. According to the DFT calculations this mode can be assigned to translations of the FA$^+$ cations [22]. However, as discussed above for MAPbBr$_3$, the intensity of this Raman band is too large for a pure translational mode and it should rather be assigned to a coupled mode involving vibrations of both the FA$^+$ cation and Pb–Br bonds.

Crystalline materials usually show an increase of wavenumbers for the majority of vibrational modes on cooling due to the shortening of chemical bonds. This behaviour was clearly observed for MAPbBr$_3$. FAPbBr$_3$ behaves in a different way, i.e., when the temperature decreases, the majority of modes exhibit an abnormal softening in the 300–260 K range. We suppose that this behaviour is observed because the expected shortening of chemical bonds on cooling is compensated by a change of the N–C–N bond angle. In case of the NH$_2$-related bands, the softening can also originate from an increase of the N–H $\cdots$ Br hydrogen bond strength. The $Pm\overline{3}m \rightarrow P4/mbm$ phase transition is evidenced as a sudden decrease of the FWHM and wavenumber of the 1369 cm$^{-1}$ band, attributed to the NH$_2$ rocking mode (see changes near 260 K in Figure 7a,c). This behaviour indicates that the phase transition is associated with an increase of the N–H $\cdots$ Br hydrogen bonds strength, which lead to a slowing down of the FA$^+$ rotational motions. Since, however, the lattice modes do not show any clear change at the phase transition and the FWHM of the 1369 cm$^{-1}$ band remains very large, it is clear that the inorganic subnetwork is weakly affected by the phase transition and the FA$^+$ cations remain disordered in the $P4/mbm$ phase.

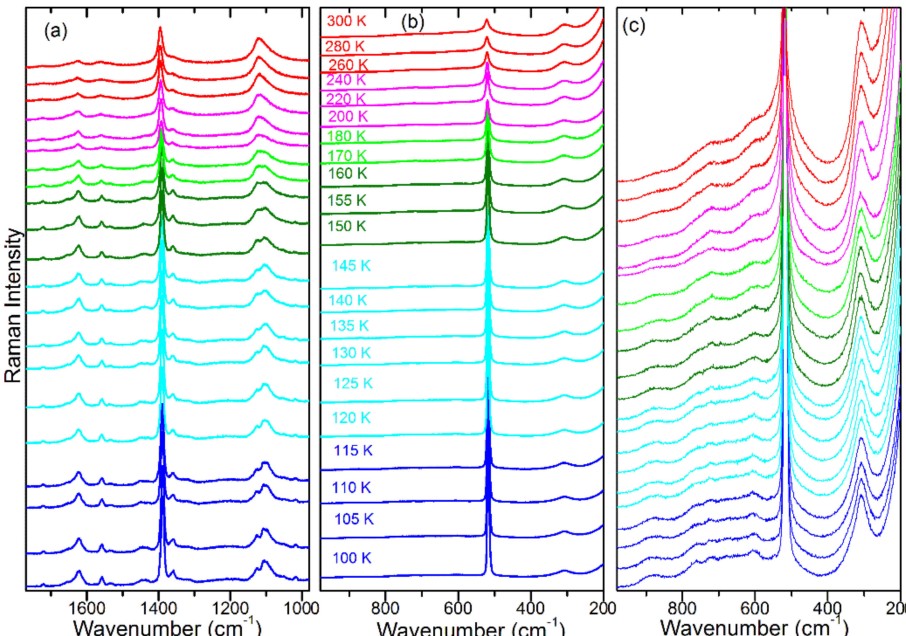

**Figure 5.** Temperature-dependent Raman spectra of FAPbBr$_3$ in the (**a**) 1770–980 cm$^{-1}$, (**b**) 980–200 cm$^{-1}$ and (**c**) 980–200 cm$^{-1}$ range (enlarged compared to panel (**b**) to show weak bands). Red colour corresponds to the $Pm\overline{3}m$ phase; magenta, green and dark green colours correspond to the $P4/mbm$ phase; and cyan and blue to the $Pnma$ phase.

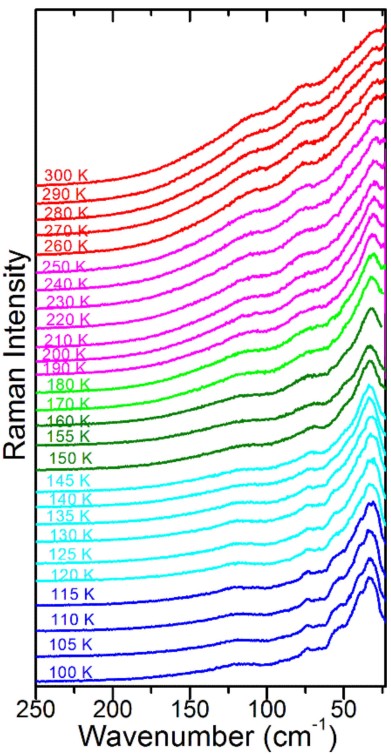

**Figure 6.** Temperature-dependent Raman spectra of FAPbBr$_3$ in the 250–22 cm$^{-1}$ range. Red colour corresponds to the *Pm$\bar{3}$m* phase; magenta, green and dark green colours correspond to the *P4/mbm* phase; and cyan and blue to the *Pnma* phase.

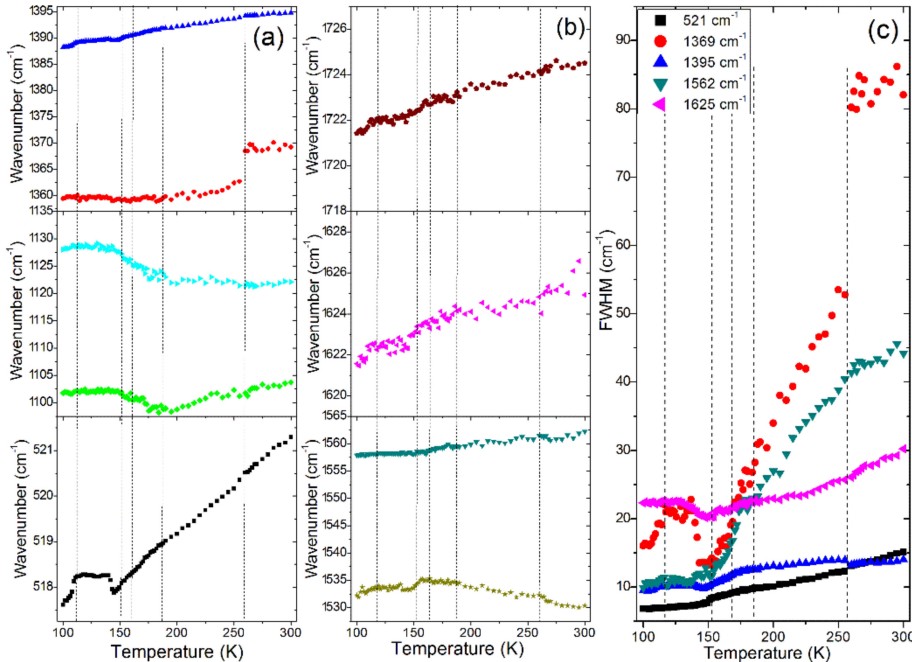

**Figure 7.** Plots of Raman (**a**,**b**) wavenumbers and (**c**) FWHM for internal modes of a FAPbBr$_3$ single crystal.

On further cooling, the Raman bands continue to narrow and shift to lower wavenumbers. However, near about 180 K, the 1122 and 1104 cm$^{-1}$ modes change their behaviour, i.e., they start to harden on cooling (Figure 7a). The FWHM values do not show any discontinuities near 180 K but a clear change of the FWHM slope vs. temperature can be

seen for the 1395 cm$^{-1}$ mode (Figure 7c). No clear changes in the lattice modes region can be noticed (Figure 6). The Raman spectra indicate, therefore, that the inorganic sublattice is hardly affected and the phase transition is related to the change of FA$^+$ dynamics. This conclusion is consistent with X-ray diffraction that could not reveal any decrease of the crystal symmetry, in spite of a very clear thermal anomaly.

On further cooling, we do not observe any clear anomalies that could indicate the onset of the phase transition observed in the thermal studies at 161 K. However, very clear anomalies are observed near 150 K, which should be attributed to the crystallographically resolved $P4/mbm \rightarrow Pnma$ phase transition. First, the 1530, 1122 and 1104 cm$^{-1}$ modes exhibit a change of wavenumber slopes vs. temperature (Figure 7a). Second, the FWHM of some modes exhibit abnormal increase (see for instance 1625, 1395 and 1369 cm$^{-1}$ modes, Figure 7c). This behaviour can be attributed to a very small splitting of these modes below 150 K, not resolved in our experiment. Third, the mode near 70 cm$^{-1}$ narrows and a new lattice mode appears at 53 cm$^{-1}$ (Figures 6 and 7). The observed changes are consistent with a lowering of the crystal symmetry and a further slowing down of the FA$^+$ dynamics. It can be noticed, however, that in contrast to MAPbBr$_3$, which showed a very strong narrowing and splitting of bands in the *Pnma* phase, the Raman bands of FAPbBr$_3$ remain broad and the changes in the lattice modes region are very weak. This proves that the distortion of the *Pnma* phase of FAPbBr$_3$ is weak and FA$^+$ cations remain disordered.

According to X-ray diffraction data, FAPbBr$_3$ exhibits one more phase transition at 118 K, not resolved crystallographically [27]. It seems that the only indication of this transition in the Raman spectra is the behaviour of the 521 cm$^{-1}$ band, which exhibits nearly no shift in the 145–115 K range but softens significantly below 115 K (Figure 7a). The temperature of this anomaly is lower by a few degrees compared to the thermal anomaly, probably due to the laser heating of the sample. Since this band corresponds to the N–C–N bending vibration, the Raman spectra suggest that the phase transition affects the N–C–N bond angle due to some change in the dynamics of FA$^+$ cations.

As discussed above for MAPbBr$_3$, the structural phase transitions are expected to affect the optoelectronic properties of FAPbBr$_3$ and the strength of this effect should depend on the magnitude of the lattice distortion and change in the molecular dynamics. Our Raman data show that although the temperature-dependent changes are much weaker for FAPbBr$_3$ than for MAPbBr$_3$, the most pronounced changes in the FWHM and shifts of the Raman bands are also observed at the tetragonal to orthorhombic phase transition near 150 K. Similar to MAPbBr$_3$, this phase transition leads to a blue shift of the PL band [31].

## 4. Conclusions

We performed temperature-dependent studies of MAPbBr$_3$ and FAPbBr$_3$ single crystals, which helped to provide a deeper insight into the phase transition mechanisms. In particular, the Raman data provided strong evidence that the intermediate phase of MAPbBr$_3$, stable in the 153–144 K range, was disordered and its structure was similar to the structure of the higher-temperature $I4/mcm$ phase. The Raman studies of FAPbBr$_3$ showed that the lattice modes exhibited clear anomalies only at the two phase transitions that could also be crystallographically resolved ($Pm\bar{3}m \rightarrow P4/mbm$ at 266 K and $P4/mbm \rightarrow Pnma$ at 156 K). However, the internal modes region could also show anomalies for two of three crystallographically unresolved phase transitions, observed in the thermal studies at 182 and 120 K. These results confirm the usefulness and the high sensitivity of vibrational spectroscopy in studies of the phase transitions related to the dynamical changes of organic cations.

**Supplementary Materials:** The following supporting information can be downloaded at: https://www.mdpi.com/article/10.3390/solids3010008/s1, Table S1: Raman wavenumbers (in cm$^{-1}$) of MAPbBr3 together with the proposed assignment; Table S2: Raman wavenumbers (in cm$^{-1}$) of FAPbBr3 together with the proposed assignment.

**Author Contributions:** Conceptualization, M.M.; methodology, M.M. and M.P.; validation, M.M. and M.P.; formal analysis, M.P.; investigation, M.P.; resources, M.M.; data curation, M.M. and M.P.; writing—original draft preparation, M.M. and M.P.; writing—review and editing, M.M. and M.P.; visualization, M.M. and M.P.; supervision, M.M.; project administration, M.M.; funding acquisition, M.M. All authors have read and agreed to the published version of the manuscript.

**Funding:** This research received no external funding.

**Data Availability Statement:** Not applicable.

**Conflicts of Interest:** The authors declare no conflict of interest.

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
