# Peer review of "Temperature-Dependent Raman Studies of FAPbBr3 and MAPbBr3 Perovskites: Effect of Phase Transitions on Molecular Dynamics and Lattice Distortion"

_solids, doi:10.3390/solids3010008_

Round 1
Reviewer 1 Report
In this work, MÄ…czka et al. performed temperature-dependent studies of MAPbBr3 and FAPbBr3 single crystals, which helped to provide deeper insight into the phase transition mechanisms. Especially, they utilized Raman spectroscopy to detect phase transitions of perovskite under various temperature conditions. Although there are several minor comments, I believe this paper has enough data and explanation to convince their key points. I recommend to accept this paper in Solids after addressing following comments.
- Authors systematically observed the phase transitions of perovskite, but there is a lack of information to show the connection between their results and applications, such as solar cells or light-emitting diodes which were mentioned at the introduction section. Please corelate the results with the applications in the manuscript for enhancing readership of this paper.
- line 53, abbreviation (NLO) should be defined.
- Although the tile has a ‘mechanisms’, I can’t find the phase transition mechanism of perovskite phase transition. I only can find phase transition information to support a mechanism. In this case, authors may need to change their title. If I was wrong, please point out the phase transition mechanism in the manuscript.
Author Response
Reviewer 1: In this work, MÄ…czka et al. performed temperature-dependent studies of MAPbBr3 and FAPbBr3 single crystals, which helped to provide deeper insight into the phase transition mechanisms. Especially, they utilized Raman spectroscopy to detect phase transitions of perovskite under various temperature conditions. Although there are several minor comments, I believe this paper has enough data and explanation to convince their key points. I recommend to accept this paper in Solids after addressing following comments.
- Authors systematically observed the phase transitions of perovskite, but there is a lack of information to show the connection between their results and applications, such as solar cells or light-emitting diodes which were mentioned at the introduction section. Please corelate the results with the applications in the manuscript for enhancing readership of this paper.
AUTHORS: We have added information on effect of the observed phase transitions on applications (see pages 6 and 8).
Reviewer 1: line 53, abbreviation (NLO) should be defined.
AUTHORS: We have defined the NLO abbreviation.
Reviewer 1: Although the tile has a ‘mechanisms’, I can’t find the phase transition mechanism of perovskite phase transition. I only can find phase transition information to support a mechanism. In this case, authors may need to change their title. If I was wrong, please point out the phase transition mechanism in the manuscript.
AUTHORS: We have modified the title and added on page 5 that the phase transition in MAPbBr3 observed at 148 K has an order-disorder mechanism.
Reviewer 2 Report
In this paper the authors present a detailed experimental investigation on the Raman spectra in two different hybrid perovskite samples of interest for photovoltaics and light-emitters. The investigated spectra range is large, larger than what can be found in literature. The fine tuning of temperature allows to detect the several changes of the crystalline structure. My overall opinion is highly positive.
I have only few requests that in my opinion will improve the manuscript:
1) I think it will be relevant to add few details on the crystal size. In fact I suppose the authors performed micro-Raman spectra and therefore it is important to comment on sample inhomogeneity if any effect was observed ( inclusions of tetragonal phase in orthorombic phase, etc.)
2) The author should add the experimental resolution being some bands quite broadened.
3) In fig.2 it is not evident the band at 47 cm -1 at 155 K. I would say that it is evident and narrow from 148 K where the authors attribute another phase transition.
Author Response
Reviewer 2: In this paper the authors present a detailed experimental investigation on the Raman spectra in two different hybrid perovskite samples of interest for photovoltaics and light-emitters. The investigated spectra range is large, larger than what can be found in literature. The fine tuning of temperature allows to detect the several changes of the crystalline structure. My overall opinion is highly positive.
I have only few requests that in my opinion will improve the manuscript:
1) I think it will be relevant to add few details on the crystal size. In fact I suppose the authors performed micro-Raman spectra and therefore it is important to comment on sample inhomogeneity if any effect was observed ( inclusions of tetragonal phase in orthorombic phase, etc.)
AUTHORS: We would like to thank the reviewer for good evaluation of our results. We have added information on size of the grown crystals, crystals used in the Raman experiment and lack of sample inhomogeneity.
Reviewer 2: The author should add the experimental resolution being some bands quite broadened.
AUTHORS: The spectral resolution was given in the original manuscript in the Experimental Section (2 cm-1). We have added, however, additional information (the following sentence was introduced “A 20x0.4 microscope magnification lens was used, size of the studied crystals was less than one mm and the laser spot diameter was about 0.75 mm.”) Regarding broadening of Raman bands of FAPbBr3 or MAPbBr3 above 150 K, it is not related to poor resolution since as you can notice Raman bands of MAPbBr3 below 150 K are very narrow (Figure 3c shows that FWHM values of some bands are near 3 cm-1). The broadening observed for FAPbBr3 or MAPbBr3 above 150 K is due to increased dynamics of the organic cations.
Reviewer 2: In fig.2 it is not evident the band at 47 cm -1 at 155 K. I would say that it is evident and narrow from 148 K where the authors attribute another phase transition.
AUTHORS: The band at 47 cm-1 is seen at 150 and 152 K as a weak band and at 148 K a band is seen at 43 cm-1, not at 47 cm-1. To avoid confusion regarding the discussed Raman bands, we have modified Figure 2 by adding arrows, which indicate onset of bands at 47 and 43 cm-1 appearing due to the phase transitions. Caption for Figure 2 was also corrected.